# Real-Time Cuffless Continuous Blood Pressure Estimation Using Deep Learning Model

**DOI:** 10.3390/s20195606

**Published:** 2020-09-30

**Authors:** Yung-Hui Li, Latifa Nabila Harfiya, Kartika Purwandari, Yue-Der Lin

**Affiliations:** 1Department of Computer Science and Information Engineering, National Central University, Taoyuan 32001, Taiwan; yunghui@csie.ncu.edu.tw (Y.-H.L.); latifanaharfiya@g.ncu.edu.tw (L.N.H.); kartikap9393@g.ncu.edu.tw (K.P.); 2Department of Automatic Control Engineering, Feng Chia University, Taichung 40724, Taiwan

**Keywords:** blood pressure (BP), electrocardiogram (ECG), photoplethysmogram (PPG), long short-term memory (LSTM), bidirectional LSTM, deep LSTM, regression

## Abstract

Blood pressure monitoring is one avenue to monitor people’s health conditions. Early detection of abnormal blood pressure can help patients to get early treatment and reduce mortality associated with cardiovascular diseases. Therefore, it is very valuable to have a mechanism to perform real-time monitoring for blood pressure changes in patients. In this paper, we propose deep learning regression models using an electrocardiogram (ECG) and photoplethysmogram (PPG) for the real-time estimation of systolic blood pressure (SBP) and diastolic blood pressure (DBP) values. We use a bidirectional layer of long short-term memory (LSTM) as the first layer and add a residual connection inside each of the following layers of the LSTMs. We also perform experiments to compare the performance between the traditional machine learning methods, another existing deep learning model, and the proposed deep learning models using the dataset of Physionet’s multiparameter intelligent monitoring in intensive care II (MIMIC II) as the source of ECG and PPG signals as well as the arterial blood pressure (ABP) signal. The results show that the proposed model outperforms the existing methods and is able to achieve accurate estimation which is promising in order to be applied in clinical practice effectively.

## 1. Introduction

According to the World Health Organization (WHO), cardiovascular diseases (CVDs) are causing great loss of life on a global scale. Most of the population die annually from CVDs than from some other cause. As reported in 2017, an estimated 17.9 million people died from CVDs in the previous year which represents 31% of all global deaths. Heart attack and stroke are kinds of CVDs that contribute up to 85% of these cases. Either people with CVD or those who have a high cardiovascular risk caused by several factors including high blood pressure, obesity, or existing established disease further need early detection and prevention using counseling or medicines adequately [1].

Blood pressure (BP) is one amongst the influential physiological indicators of human health [2]. BP is closely related to cardiac function and peripheral blood vessels and is essentially used as a direct measurement for cardiovascular functions. BP changes in accordance with the process of blood transfer from heart to arteries. In every beat the heart contracts causing BP in the vessels to hit its maximum and rests in contrast causing BP in the vessels to reach its minimum, alternately. When BP hits its maximum, we refer to the value as systolic blood pressure (SBP), while the minimum is diastolic blood pressure (DBP). BP, as a matter of fact, varies over time due to numerous factors such as diet and mental state. Therefore, monitoring BP in a continuous way is necessary for accurate diagnosis and further treatment of related people [3].

The conventional cuff-based BP measurement devices have repeated measurement operation which is discontinuous in nature, with an operation interval greater than at least one minute [4]. Hence, cuff-less BP measurement utilizing related biomedical signals becomes sought-after while accurate and effective BP estimation plays a vital role in clinical practice [5]. In recent years, some methods of BP detection and evaluation have been proposed. Feature engineering methods, particularly the methods based on arterial wave propagation theory and photoplethysmogram (PPG) morphology theory, have been studied most. The former method measures pulse transit time (PTT) to predict blood pressure levels [6]. This method usually requires two physiological signals, such as electrocardiogram (ECG) and PPG signals. This approach has been explored by several past studies (for example, [7,8,9]) which verified the feasibility of the solution.

Another approach that assesses blood pressure levels by establishing a PPG morphological feature model has also been proposed [3]. This method requires a high-quality PPG signal, such as high sampling rates and sampling precision, and it is very sensitive to many kinds of noises. As the acquisition of ECG and PPG signals are popular in clinical settings and can provide more information to the regression model, this research adopts ECG and PPG simultaneously to estimate BP values.

There are so many established regression methods, such as support vector machines (SVM), linear regression, regression trees, model trees, ensemble of trees, and random forest [9,10]. The regression models for BP estimation involve the unknown parameters of the model denoted as α (algorithm-specific), the independent variables X (features from PPG or ECG) and the dependent variable Y (either SBP or DBP). In [11], the authors proposed the GA-SVR (Genetic Algorithm Support Vector Regression) BP models to estimate the SBP and DBP. In the research performed in [10], the authors estimated the SBP and DBP values using a regression tree, multiple linear regression (MLR), and SVM. A 10-fold cross-validation was applied to obtain overall BP estimation accuracy separately for all three machine learning algorithms.

Nowadays, most researchers apply deep neural networks because they allow them to have large amounts of labeled input data and be capable of modeling extremely complex and non-linear relationships between inputs and outputs. For BP estimation, the output layer of the network often consists of two neurons, one for SBP and the other for DBP [7,12,13,14]. The metric used for the performance evaluation of the model was mean absolute error (MAE) [7,12,13,14,15]. Based on [14], the authors proposed a regression model based on the deep belief-network (DBN)-deep neural network (DNN) to learn about the complex nonlinear relationship amidst the generated feature vectors obtained from the oscillometric wave and the observed blood pressures.

As mentioned in the previous study [16], ECG and PPG are the signals that can be used to retrieve the important information for BP estimation. The temporal dependency between the patterns of ECG, PPG, and arterial blood pressure (ABP) signal can be seen in Figure 1. The ECG signal consists of five wave segments P, Q, R, S, and T, while the waveform of the PPG signal describes the systole and diastole of the cardiac cycle. Using these signals, we can derive the PTT value by measuring the time difference from the R peak of the ECG signal to the maximum slope on the PPG signal. On top of that, it is possible to derive other parameters, such as R to R peak interval in ECG, the amplitude of systolic peak, dicrotic notch, and second peaks, and the time interval between beats in PPG so that it is possible to extract more features.

In practical conditions, the term PTT refers to the travel time between aortic valve opening and arrival of the blood flow to the distal location. When the time is measured relative to the ECG QRS complex then it is generally used to define the term pulse arrival time (PAT), an interchangeably measure of PTT. Despite that, both timings could implicate poor correlation due to the variability of the pre-ejection period (PEP), which is from the related ECG QRS complex to the aortic valve opening [17]. This article follows the usage in previous literature [3,6,16], and the term PTT is also adopted to avoid confusion.

In this work, we propose the deep long short-term memory (LSTM) networks to perform real-time BP estimation utilizing the features extracted from the ECG and PPG signals. The theories behind the BP estimation correlated features and methods introduced in Section 2 and Section 3 give details about the utilized methods and the overall framework. Details about the dataset and the environmental setup are presented in Section 4. Our experiment results are reported in Section 5 along with the discussion. Lastly, the paper is concluded in Section 6.

## 2. Background

BP is knowingly correlated to pulse wave velocity (PWV) [16]. The artery is assumed to be a cylindrical elastic tube with a radius of r (in unit m) and thickness of h (in unit m). Based on the Moens–Korteweg equation [18,19], we have the pulse wave velocity (PWV, in unit m/s) as follows:(1)$$\mathrm{PWV}=\sqrt{\frac{E\cdot h}{2\cdot r\cdot \rho }}$$
where E denotes the elastic modulus of arterial wall (in unit mmHg), and ρ represents the density of blood in the artery (in unit kg/m^3^).

As the BP increases, the velocity of the pulse wave traveling in the vessels is increasing as well. If the pulse wave is detected by two sensors apart from a distance of D on the artery, then the pulse wave transit time (PTT, in unit s) can be derived by
(2)$$\mathrm{PTT}=\frac{D}{\mathrm{PWV}}=D\cdot \sqrt{\frac{2\cdot r\cdot \rho }{E\cdot h}}$$

It is assumed the elastic modulus of the arterial wall E is a constant in Equation (2). In fact, the value of E in the artery is verified to be exponentially increased with the blood pressure P (in unit mmHg), especially for central artery near the heart, and can be represented by [20]:(3)$$E\left(P\right)=E_{0}\cdot e^{\alpha \cdot P}$$
where $E_{0}$ denotes the elastic modulus at 0 mmHg and α is a real-valued parameter larger than zero that is closely related to arterial stiffness. The value of α is greater for the stiffer arteries.

Substitute Equation (3) into (2) and the equation can be manipulated to be
(4)$$P=-\frac{2}{\alpha }\cdot \ln\left(\mathrm{PTT}\right)+\frac{1}{\alpha }\cdot ln\left(\frac{2\cdot r\cdot \rho }{E_{0}\cdot h}\cdot D^{2}\right)$$

It can be observed that there exists a nonlinear relationship between blood pressure P and PTT. This equation implies that the BP value can be estimated from PTT which has been commonly used as an indicator to indirectly estimate BP continuously and cufflessly in previous studies [21]. Apart from its definition as the time taken by the arterial pulse propagating from the heart to a peripheral site, PTT can be calculated as the time interval between the ECG peak with a maximum slope point (first derivative) of PPG as well [22]. However, since there exist several parameters that are highly dependent on the personal arterial characteristics, as shown in Equation (4), it also implies that the BP estimation based on PTT is a challenging task.

Furthermore, the PTT-based BP estimation technique has not been widely accepted yet for cuffless and continuous BP monitoring because its estimation accuracy is limited for clinical uses [23]. The reason can be obtained indirectly from Equation (4), where it can be observed that the relationship between BP and PTT is not only nonlinear but also that some parameters are closely related to personal arterial characteristics. These issues make PTT-based BP estimation unsatisfactory for a clinical requirement. In order to cover the information about personal arterial characteristics, some other parameters are also included in this research (see Section 3).

## 3. Methodology

We propose a blood pressure estimation algorithm based on a deep learning model that contains BiLSTM (Bidirectional Long Short-Term Memory) in the first layer and followed by n layers of LSTM. Specifically, we used residual connection for each LSTM layer with the input of seven features listed on Table 1. The features are generated from the ECG signal or/and PPG signals. Some parameters that are applied to generate features from the PPG signal are shown in Figure 2.

### 3.1. Preprocessing

#### 3.1.1. Low-Frequency Removal by Fourier Transform

The discrete Fourier transform (DFT) is the method that can be used for converting time-domain signals into the frequency domain to obtain frequency coefficients for discrete-time sequences [24]. Let x[n], 0≤n≤N−1, represent the input signal; the DFT of x[n] is denoted as X[k], 0≤k≤N−1. The low-frequency artifact can be reduced by turning off the low-frequency component as follows:(5)$$X_{r}\left[k\right]=\{\begin{array}{cc} X\left[k\right] & k\geq k_{c}\\ 0 & otherwise \end{array}$$
where $k_{c}$ is the upper limit frequency index of low-frequency artifact. Let $f_{s}$ denote the sampling frequency of the signal (in Hz) and N the length of DFT; then, the relationship between the practical frequency f and the frequency index k of DFT is $f=\left(k\cdot f_{s}\right)/N$ Hz [25]. The sampling frequency of signals adopted for this research is 125 Hz, and the DFT length N is equal to 4096. The signal can then be restored with the inverse DFT, as follows:(6)$$x_{r}\left[n\right]=\frac{1}{N}\sum_{k=0}^{N-1}X_{r}\left[k\right]\cdot e^{j\frac{2\pi n\cdot k}{N}},\;0\leq n\leq N-1.$$

The value of $k_{c}$ is derived by $\left\lfloor \frac{N\cdot f_{c}}{f_{s}}\right\rfloor$, where $f_{c}$ is the upper limit for the low-frequency artifact and the symbol ⌊·⌋ represents the operator that rounds the value to the nearest integer toward the direction of negative infinity. The sampling frequency of signals adopted for this research is 125 Hz, and the DFT length N is equal to 4096. The upper limit for the low-frequency artifact is 0.1 Hz in this research. From these parameters, the value of $k_{c}$ in Equation (5) is derived to be 3.

#### 3.1.2. Normalization

We normalize the amplitude of ECG and PPG signals to the range [0, 1] to ease and robustify the process of comparisons and analysis. The unit of ECG and PPG signals’ amplitude is arbitrary, and the feature extracted from it can depend on the amplification/scaling of individual recordings. Thus, normalization is done to make sure that the value extracted is meaningful. This process is done by dividing the subtraction of each sample i in the signal x and the minimum value of the corresponding signal with the subtraction of the maximum value and the minimum value of the corresponding signal, as follows:(7)$$x_{i}^{\prime }=\frac{x_{i}-\min\left(x\right)}{\max\left(x\right)-\min\left(x\right)}$$

### 3.2. Features Extraction

We follow the procedure described in [26] for features extraction. There are seven features to be used as the input to the deep learning models. To extract those features, first we do R peaks’ detection on the whole ECG signal using the Pan–Tompkins algorithm [27]. Based on the fiducial R peaks, the other four parameters including foot, systolic peak, dicrotic notch, and second peak (as indicated in Figure 2) are also detected on the whole PPG signal. After that, we specify the window size of a cycle. Here, we define a “cycle” as a time window that is always started from R peak, followed by systolic peak, next R peak, and ended by the following R peak. While performing sliding window method, with a window size less than or equal to 200 data points, we extract the seven features for each window. Thus, every cycle that does not meet the criteria is skipped as we consider it as abnormal.

All of the features are listed in Table 1 along with the calculation and the unit for each feature. Feature number 1 (PTT) is obtained by computing the distance between the ECG R peak and the PPG maximum slope within half a cycle. Feature number 2 (HR) is obtained from both R peaks in a cycle while the rest of the features (RI, ST, UT, SV, DV) are obtained within the range of the first foot to the second foot of a cycle. We pick the definition of foot, denoted as $tf_{n}$, as the minimum of the waveform in the region. The maximum of the waveform following the foot is called the systolic peak, denoted as $tp_{n}$. We take the minimum of the subtraction between the signal and the straight line going from the systolic peak to the foot as $tn_{n}$ which denotes the dicrotic notch. Lastly, the second peak is defined by the minimum of the second derivative of the waveform following the dicrotic notch, which is denoted as $td_{n}$ [28].

### 3.3. Multiple Linear Regression (MLR)

Linear regression analysis is carried out to make predictions about the values of the dependent variable, Y, based on one explanatory variable. MLR, as an extension to that, uses a set of p explanatory variables ($x_{1},x_{2},\ldots ,x_{p}$) instead of one. With the assumption that Y is directly related to a linear combination of the x, the relationship between the dependent and the explanatory variables is modeled as
(8)$$Y_{i}=\beta_{0}+\beta_{1}x_{1i}+\beta_{2}x_{2i}+\ldots +\beta_{p}x_{\mathrm{pi}}+\epsilon_{i}Y_{i}=\beta_{0}+\beta_{1}x_{1i}+\beta_{2}x_{2i}+\ldots +\beta_{p}x_{\mathrm{pi}}+\epsilon_{i}\;Y_{i}=\;\beta_{0}+\;\beta_{1}\;x_{1i}+\beta_{2}\;x_{2i}+\ldots \;+\;\beta_{p}\;x_{pi}+\epsilon_{i}$$
where *i* denotes the observational unit of n from the dependent variable, $\beta_{0}$ is the constant term, $\beta_{1}$ to $\beta_{p}$ are the coefficients relating the p explanatory variables to the variables of interest, and ϵ is the error term (which is also known as residual) of the model [29].

### 3.4. Regression Ensemble

Some methods built for solving regression problems are mentioned as weak learners due to their poor performance. These learners usually combine complex models to improve performance. The ensemble method must comprise of a different character of single learners to reduce the prediction error in regression. For instance if one single learner can have a high bias but low variance then the combination pair should be a learner with strength in reducing the bias [30]. Taxonomically, there are several ensemble techniques. A well-known category, called data resampling, is trying to obtain a unique learner by generating different training sets. The following three kinds of algorithm that aim at combining weak learners belong to such category [30]:Bagging: The base learners are trained on the resampled training set independently from each other in parallel and are combined by following a deterministic averaging process. It is a combination of bootstrapping and aggregation methods to form an ensemble model.Boosting: The base models learn sequentially in an adaptive way which combines them by following a deterministic strategy. It starts from learning on the whole data set by giving an equal weight to each observation, whereas the following learns on a training set based on the performance of the earlier. The higher weight is given to the observation corresponding to the worse performance.Stacking: It consists of two stages that firstly generate a new dataset from the output of parallelly-trained base models. Then, the dataset is used for meta-algorithm learning to produce the final output.

### 3.5. Deep LSTM

LSTM is an exceptional kind of recurrent neural network (RNN) that is introduced as one breakthrough in handling long-term dependencies by resolving the vanishing gradient problem. A slight nuance between the architecture of RNN and standard LSTM is the hidden layer or the so-called cell [31]. The core point behind the LSTM success is the memory cell that can preserve its state over time and its nonlinear gating units that thoroughly regulate the information flow [32], which is shown in Figure 3.

Fundamentally at time t, the LSTM cell is composed of a layer input $x_{t}$ and a layer output $h_{t}$. The complicated cell also carries the cell input state $\tilde{C_{t}}$, the cell output state $C_{t}$, and the output state of the prior cell $C_{t-1}$. This information is learned during the training and consequently used for updating parameters. The cell state is protected and regulated by three gates consist of forget gate $f_{t}$, input gate $i_{t}$, and output gate $o_{t}$. As illustrated in the same figure, the gates are composed out of a sigmoid σ as the activation function. Due to its gate structure in cells, LSTM can deal with long-term dependencies to allow useful information to pass through and to let the redundant information be removed from the LSTM network [33].

One layer of LSTM is composed of multiple memory cells. Given T number of cells of time steps, the final output of an LSTM layer is a vector comprises of all the cell outputs $h_{t}$, represented by $Y_{T}=\left[h_{T-n},\ldots ,h_{T-1}\right]$. In this paper, when we take the BP estimation problem as an example, only the last element of the output vector, $h_{T-1}$, is what we want to predict. Hence, the estimated blood pressure value $\hat{x}$, for the next iteration time T, is namely ${\hat{x}}_{T}=h_{T-1}$.

The depth of network remarkably contributes to the success of solving many filed tasks. Essentially, we can see the deep network as a processing pipeline; that in each layer a part of the task is being solved before conveying it until to the final layer to provide the output [34]. With a hypothesis that increasing the depth of the network provides a more efficient solution in solving the long-term dependency problem of sequential data [35], we present four different deep LSTM models by stacking several LSTM layers to yield the performance at estimating the blood pressure.

### 3.6. BiLSTM

Bidirectional LSTM (BiLSTM) connects the two hidden layers of LSTM to the output layer. Having two LSTM as one layer in the application prompts enhancing the learning long-term dependency and, along these lines, it subsequently will improve the model performance [36]. A prior study proved that the bidirectional networks are significantly better than the standard ones in many fields [37], including the BP estimation case as well [7]. The structure of an unfolded BiLSTM layer which contains a forward LSTM layer and a backward LSTM layer is illustrated in Figure 4.

As the forward LSTM layer output sequence $\vec{h}$, is obtained in a common way as the unidirectional one, the backward LSTM layer output sequence $\overset{\leftarrow }{h}$, is calculated using the reversed inputs from time t−1 to t−n. These output sequences then fed to σ function to combine them into an output vector $y_{t}$ [33]. Similar to the LSTM layer, the final output of a BiLSTM layer can be represented by a vector, $Y_{t}=$[$y_{t-n},\ldots ,y_{t-1}$], in which the last element, $y_{t-1}$, is the estimated blood pressure for the next iteration.

### 3.7. Residual Connection

Empirical evidence found that residual connection can improve performance on long-term dependency task significantly [38]. As shown in Figure 5, the formulation of F(x)+x can be perceived by feedforward neural networks with shortcut connections which simply perform identity mapping as their further outputs are inserted to the outputs of the stacked layers.

The building block of residual is like this formula:(9)$$Y=F\left(x,\left\{W_{i}\right\}\right)+x$$
where x and Y are the input and output vectors of the layers considered. The function $F\left(x,\left\{W_{i}\right\}\right)$ represents the residual mapping to be learned. For the example in Figure 5 that has three layers, $F=W_{2}\sigma \left(W_{1}x\right)$ in which σ denotes ReLU [39] and the biases are omitted for simplifying notations.

The operation *F*(*x*) +
*x* is performed by a shortcut connection and element-wise addition. We adopt the second nonlinearity after the addition (i.e., σ(y), see Figure 5) [40]. In this paper, we present four deep LSTM models, with the three proposed models using the residual connection while the other does not. By doing this, we can see the impact of residual connection at improving the deep LSTM model for estimating blood pressure.

### 3.8. Overall Framework

The overall flow diagram of the proposed methodology is presented in Figure 6, which is summarized in the following steps:Extract the ECG, the PPG, and the ABP signals.Preprocess the ECG and PPG signals, which primarily includes the removal of baseline wander and motion artefacts.Derive waveform features from cycles of the preprocessed ECG and PPG signal and reference BPs (SBP and DBP) from the ABP signal.Train models for BP estimation using machine learning, deep learning, and the proposed model.Evaluate the estimation accuracy of SBP and DBP.

The LSTM network used in our experiment is illustrated in Figure 7. In the diagram, the initial state $C_{t-n}$ is a zero vector, c is the number of features, d is the number of hidden units, and T is the number of time series defined as the signal’s length. We feed the network with the c×T feature matrix X and set the hidden unit to 256. Thus, for each LSTM cell at the current time step t, the input layer $x_{t}$ is a set of c features.

Let $X_{T}=\left[x_{1},\;x_{2},\;\ldots ,\;x_{T}\right]$ be the network input feed and the target BP sequence is denoted as $Y_{T}=\left[y_{1},\;y_{2},\;\ldots ,\;y_{T}\right]$. We can factorize the conditional probability of $p(Y_{T}|X_{T})$ as follows:(10)$$p(Y_{T}|X_{T})=\prod_{t=1}^{T}p(y_{t}|h_{t})$$
where $h_{t}$ is a set of 256 hidden units and has been denoted as the layer output in the LSTM cell. It is generated from previous hidden state $h_{t-1}$ with the current input $x_{t}$ as follows:(11)$$h_{t}=f\left(h_{t-1},\;x_{t}\right)$$

The last $C_{t}$ of the sequence will be the final state of the corresponding LSTM layer. This state acts like a memory to preserve historic information and will be used as the initial state for the next following LSTM layer.

We propose four different LSTM models as well which are shown in Figure 8. All of the proposed models are started by one layer of BiLSTM followed by a series of stacked LSTM layers, a fully connected (FC) layer, and the regression layer as the output. We use BiLSTM in the first layer to fully capture the semantic information of the whole input signal as it takes the input sequence in its original and reverse order [41]. We then apply a residual connection in the following stacked LSTM layer to overcome the exploding or vanishing gradient problem. The residual connection allows gradients to directly flow through the network without passing a nonlinear activation function whose nonlinearity nature causes the gradients to explode or vanish. The output from the last layer of stacked LSTM is fed into the FC layer and then regresses into two scalar outputs which represent SBP and DBP values. The difference between the number of the LSTM layer and the residual connection is indicated in the performance. These models’ performances will be shown and discussed in Section 5.

## 4. Experiment

### 4.1. Dataset

The database utilized in this paper is from the multivariate intelligent monitoring in intensive care II (MIMIC II), which is an online waveform database provided by Physionet. MIMIC II contains more than 25,000 instances, indicating a patient’s record, and three biomedical signals are collected for each instance. The signals include:(1)PPG signal acquired from the sensor placed on the fingertip.(2)ABP signal is acquired by invasive probe, recorded in the unit of mmHg.(3)ECG signal (usually lead II) is recorded by an electrocardiograph.

The waveform signals are sampled at a frequency of 125 Hz. As the signals in the original MIMIC II database may be corrupted by motion or intermittent for unknown reasons, the cleaner dataset that has been prescreened from MIMIC II in the literature [3] is adopted in this paper. The dataset includes the raw ECG, PPG, and ABP signals and have already been converted into the MATLAB binary data format [42]. Our first thing to do is to extract the three signals for each record since the database consists of a cell array of matrices with each cell representing one record. In each matrix, each row corresponds to one signal channel. The statistic of the dataset based on the ABP signal is shown in Figure 9.

The ECG signal may be affected by various noises in its frequency range (0.5–150 Hz). This range contains different internal and external noises with the most common noises being the muscle artefact, baseline wander, and power line interference (PLI) which may be due to the muscle contraction, body movement, respiration, poor contact between the electrode and the subject’s skin, and so on [10].

The PPG signal’s quality naturally relies upon the area and the properties of the subject’s skin at measurement, including the individual skin structure, the blood oxygen saturation, blood flow rate, skin temperatures, and the measuring environment. These factors create a few sorts of added substance of artefact which may be contained within the signals which may influence the extraction of features and hence the overall diagnosis, particularly when the signal and its derivatives will be assessed in the algorithm. Taking a dicrotic notch (see Figure 2) as an example, this property is commonly observed from the compliant arteries in the catacrotic phase of subjects but may not appear in the signal due to some occurrences during the acquisition.

As PLI is not apparent for the ECG and PPG signals in the MIMIC II dataset, the noises that have to be removed are those low-frequency artifacts buried in the signals. To enhance the quality of the ECG and PPG signals for further analysis, we remove the low-frequency components from the signal to reduce the baseline wandering effect as well as other low-frequency artefacts. This process is done by applying Fast Fourier Transform (FFT), the fast algorithm of DFT (refer to Section 3.1.1). After removing segments that are corrupted by noises and unexpected movement of the users, there are a total of 1,113,634 records of cycle from 3000 randomly selected subjects with features including PTT, HR, RI, ST, UT, SV, and DV extracted from the ECG and PPG signals (refer to Section 3.2). We also extract the SBP and DBP values from the ABP signals as the ground truth. We further remove some parts with very high or very low BP values (e.g., SBP ≥ 180, DBP ≥ 130, SBP ≤ 80, DBP ≤ 60). The final datasets consist of 678,202 records of a cycle. In Table 2, we show statistical information about the distribution such as the standard deviation (STD) and mean as well as the ranges of the DBP, MAP, and SBP values in the datasets. We randomly select 80% of them as the training set and reserve the remaining for the testing set. So, the total number of records is 542,561 for training and 135,641 for testing. The training set and testing sets are disjointed completely.

Following our network implementation, the dataset is organized into a 7×T matrix $X_{T}$, and each row of $X_{T}$ is normalized to have zero-mean and unit variance. As mentioned in Section 3.8, T is the number of columns based on the length of the signal in each subject. The normalized features will become the input to the proposed LSTM model. The ground truth (we take only SBP and DBP values from the ABP signal) is then normalized into the range of [0, 1].

### 4.2. Environment Details

Various deep learning-based open source libraries such as Tensorflow, Keras, Theano, and Caffe have been recently provided. In this paper, we use the Deep Learning Toolbox from Matlab which provides a framework for designing and implementing deep neural networks [42]. We perform experiments using MATLAB 2019A inside the Windows 10 Enterprise computer with Intel^®^ Core™ i5 3.2 GHz processor. We have RAM of 8GB and GPU GeForce GTX 750 Ti 18GB equipped on the computer as well.

### 4.3. Error Metrics

We use three kinds of error metric calculation to measure the error from the models in the experiments, namely mean absolute error (MAE), root mean square error (RMSE), and standard deviation (STD) [43]. Let the model-predicted value be considered as a prediction result for the given input features, and the error be the distance between the observed value of BP and model-predicted value.

## 5. Results and Discussion

Following the steps in the flow diagram of the proposed methodology (see Figure 6), we present the experimental results. We compare the performance of our proposed deep LSTM models, traditional machine methods for regression problem, and the existing deep learning model to estimate the value of SBP and DBP. Next, the discussion on the analysis of experimental results is given at the end of this section.

### 5.1. Experimental Results

Our proposed models use deep residual connected LSTM layers (number of layers varies from 4 to 6), and BiLSTM is embedded at the first layer in each model (refer to Figure 8). On the bottom of the network, we append an FC layer and a regression layer. For each LSTM and BiLSTM layer, there are 256 hidden layers, with a dropout rate set to 0.2, and the activation function set to ReLU. The initial learning rate is set to 0.003 and will decrease in every 125 periods with a factor of 0.2. The maximal epochs were set to 50. The Adam Optimizer is used in the training process. Figure 10 shows the final real-time demonstration program developed in our laboratory with the subject numbered 315 as an example. Both of the SBP and DBP prediction results (presented as “Estimated”) are showing very small errors to the observed value (presented as “Practical”).

### 5.2. Discussion

There are three traditional machine learning methods, namely the linear regression, random forest [44,45] with bagging optimization, and least squares (LS) boost [46] which is a linear regression with a boosting algorithm used as a baseline in this comparison task. We conduct the experiment on 50 records of subjects which comprised of 6852 records of a cycle. We partition this data disjointly; 80% for training (5482 records) and the rest for testing (1370 records) with all the results are shown in Table 3. All of the proposed models produce better results compared to the baseline (shown as underlined and bold). The MAE, STD, and RMSE for linear regression are the highest. The performance of random forest is approaching the proposed models. In particular, the proposed models present a strong advantage over the traditional machine learning methods because they can model the mapping function between the input features and output BP by learning the temporal relations between successive frames, which enhance their prediction ability. Moreover, deep learning models can be trained with less human intervention compared with the traditional machine learning methods.

Bias and variance are the two most important features that we would like to observe for a machine learning model. Bias is how far the predicted values are from the ground truth values. If the average predicted values are far off from the ground truth values, the bias will be high. In the same table, we can find that Model 2 has the lowest bias, but the variance is a little bit high. Based on [47], this phenomenon shows that the models are somewhat accurate but inconsistent on average. As for model 4, it has the lowest variance but a higher bias. It shows that the model is consistent but inaccurate on average. This is commonly called the bias–variance tradeoff and gets into the heart of why machine learning is difficult.

We evaluate the performance of our best proposed model, Model 2, based on British Hypertension Society (BHS) standard [48] and Association for the Advancement of Medical Instrumentation (AAMI) standard [49]. According to the BHS standard, the proposed model acquires grade B and A for SBP and DBP prediction, respectively. This grading is based on the cumulative percentage errors under particular thresholds, as shown in Table 4. In the same table, the comparison between our model results on mean error (ME) and STD with the AAMI criterion is presented. The AAMI recommends a test on more than 85 subjects with ME and STD under 5 mmHg and 8 mmHg, respectively. Our model is trained and tested on a larger population of subjects and the overall result satisfy the standard margin. However, the STD value of the SBP prediction is out of the margin which might be due to the uniqueness of the human body, especially the fact that our model tested on a very large set which is selected randomly. The distribution of the absolute error of both SBP and DBP prediction is demonstrated in Figure 11.

In Table 5, we compare our best model with an existing deep learning model proposed in [7]. This model embodies two levels of hierarchy which consist of artificial neural networks (ANN) to extract the features in the lower level and LSTM to learn the temporal relations amongst those features in the upper level. We use our preprocessed data to conduct a fair comparison with the number of records mentioned in Section 4. The results show that our model outperforms this model despite the variance of the length of the feature vector sequence, which is 10 and 32. We also compare our model with a deep neural network (DNN) which consists of five FC layers. This DNN model is created with the same structure of the model 2. We replace the BiLSTM and the LSTMs in the model with ANN to test the importance of the memory for the BP estimation task. The number of neurons in the first four layers is set to 256 and we apply residual connection for every layer. The result shows that applying memory cell into the model reduce the overall error significantly.

In statistical learning theory, tuning model complexity precisely is one of accomplishment considering it as a subtle idea. Essentially, models with higher complexity require a greater number of training datasets. Thus, the variance in the learned models is going to be much larger for the models with higher complexity than the simpler ones. We ought to use a definite measure of prediction error and explore different levels of model complexity. Accordingly, we can determine the complexity level that performs with the lowest overall error. The fundamental thing to this procedure is the selection of the accurate error measures as wrong error metric is prone to misleading the research direction [47,50].

## 6. Conclusions and Future Work

In this paper, a deep learning model based on the ECG and PPG signals for the continuous estimation of SBP and DBP has been implemented. We perform the experiments to compare the accuracy between the traditional machine learning methods and the existing deep learning-based method. The experimental results show that the proposed models can reduce the value of MAE, STD, and RMSE for both SBP and DBP prediction. It shows that our model is state-of-the-art compared to the baseline. Furthermore, the overall performance for DBP estimation passes the BHS and AAMI standard while the SBP estimation passes the BHS standard only.

Nonetheless, our proposed model relies on ECG and PPG signals which have some limitations. Despite our motivation to build a model for BP monitoring in continuous way, the acquisition of a long-term ECG signal itself can be cumbersome. Moreover, some PPG signals did not have a dicrotic notch at a certain moment. Therefore, features based on dicrotic notch may not be available at all times. The dataset we used in this study acquires PPG from finger-tip sensors that are mostly used in clinical settings only. For more efficient BP monitoring, a wearable device with a PPG sensor on a wrist can alternatively be used. However, the signal contains a lot more noise compared with the PPG obtained from finger-tip sensor. In the future, we will try to exclude features based on parameters such as a dicrotic notch to estimate the blood pressure. Another direction is that we may try to use the current features together with the derivatives of them (∆PTT, ∆HR, ….) to see if the accuracy can be further enhanced.

## Figures and Tables

**Figure 1 sensors-20-05606-f001:**
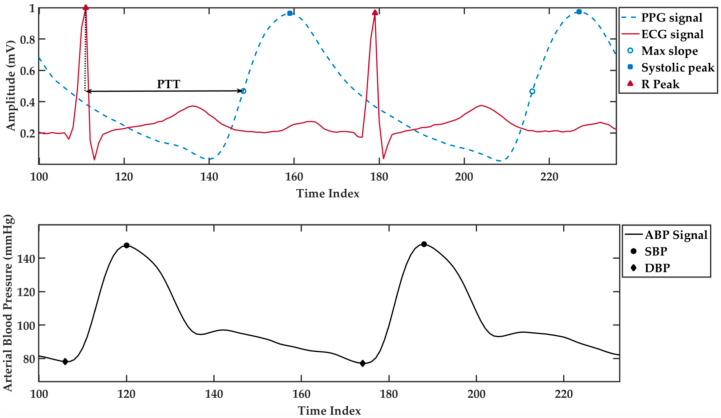
The temporal dependency between the patterns of an electrocardiogram (ECG), photoplethysmogram (PPG), and arterial blood pressure (ABP) signals.

**Figure 2 sensors-20-05606-f002:**
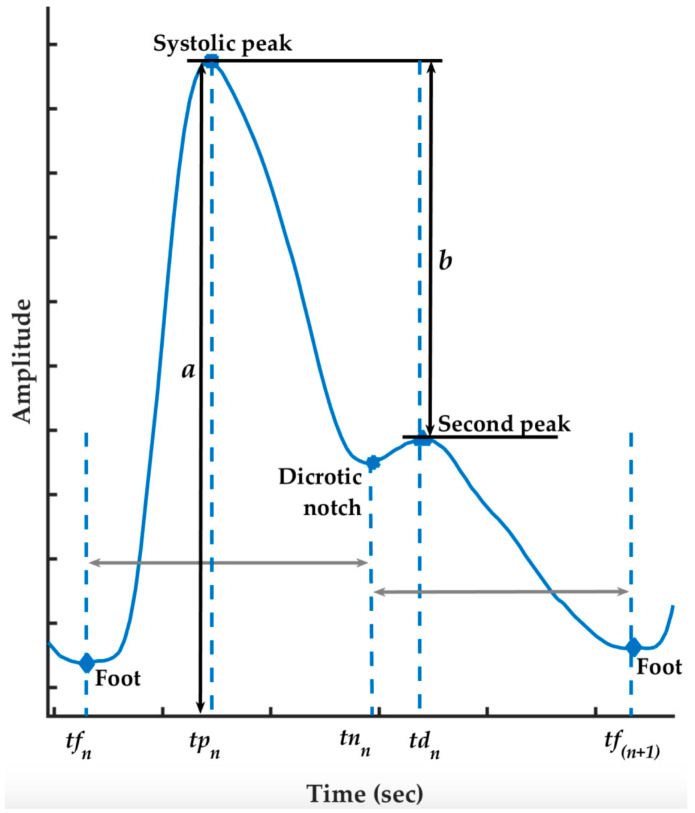
Illustration of PPG parameters.

**Figure 3 sensors-20-05606-f003:**
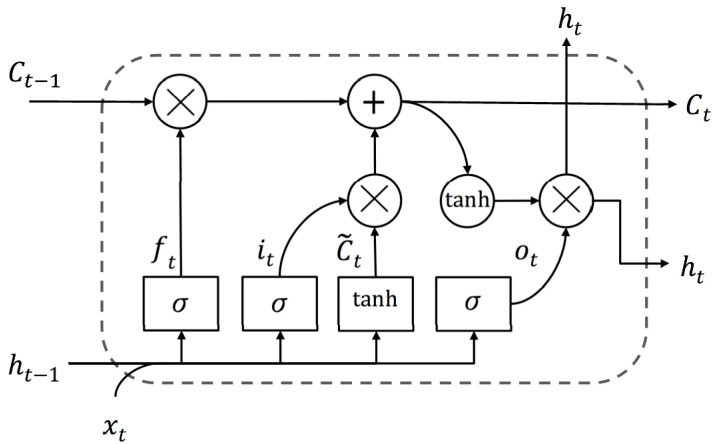
Long short-term memory (LSTM) cell with its gates presented as rectangles.

**Figure 4 sensors-20-05606-f004:**
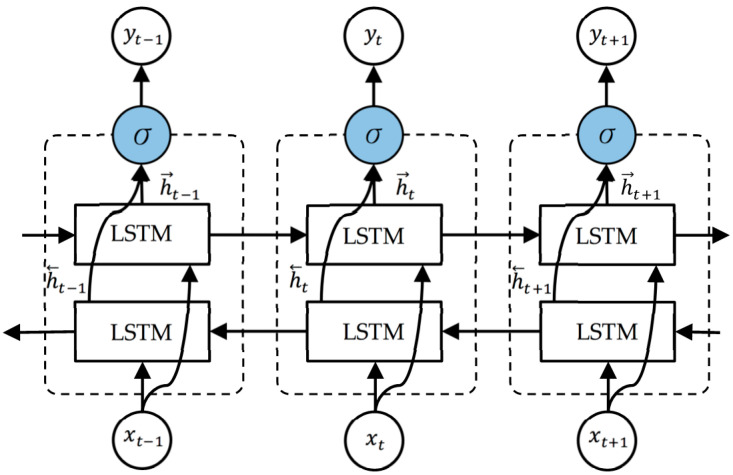
The unfolded architecture of Bidirectional LSTM (BiLSTM) with three consecutive steps.

**Figure 5 sensors-20-05606-f005:**
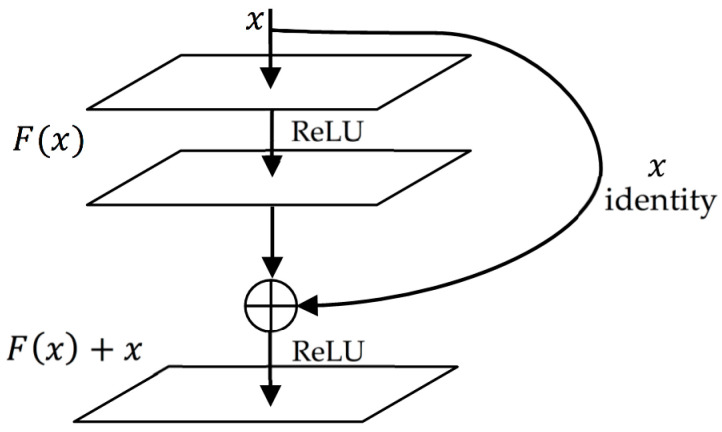
Residual connection.

**Figure 6 sensors-20-05606-f006:**
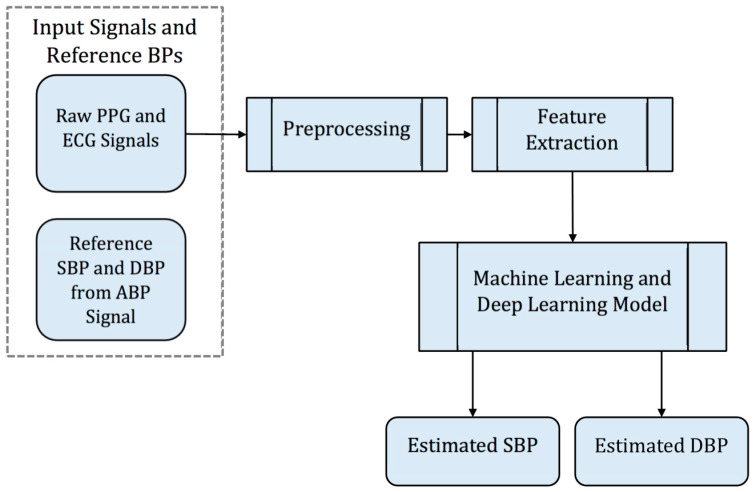
Flow diagram of the proposed methodology.

**Figure 7 sensors-20-05606-f007:**
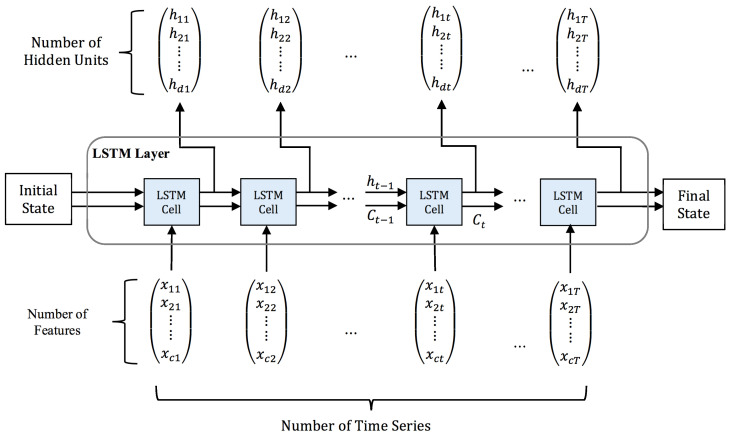
Illustration of the LSTM network used in the experiment.

**Figure 8 sensors-20-05606-f008:**
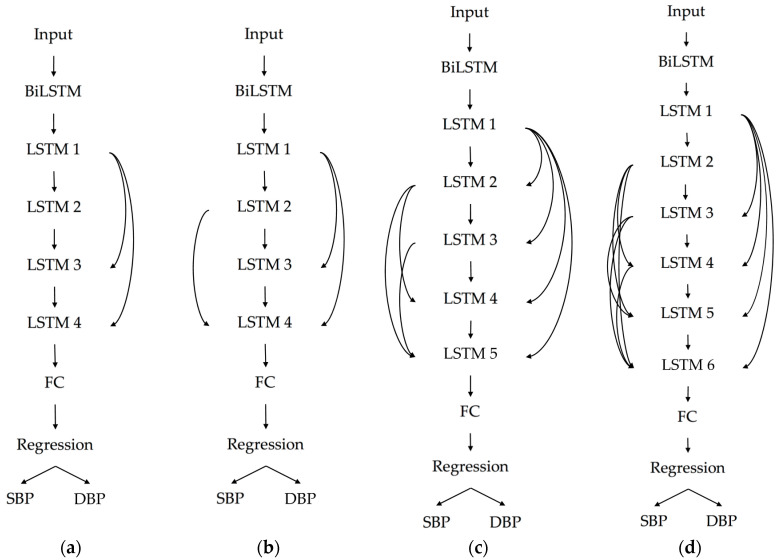
The proposed models: (**a**) Model 1, (**b**) Model 2, (**c**) Model 3, and (**d**) Model 4.

**Figure 9 sensors-20-05606-f009:**
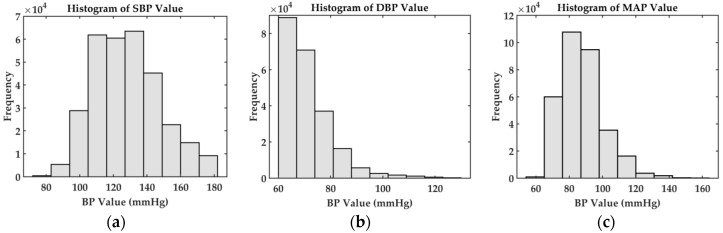
Statistics of the datasets used in the experiments. (**a**) Histogram of systolic blood pressure (SBP) values, (**b**) histogram of diastolic blood pressure (DBP values, and (**c**) histogram of mean arterial pressure (MAP) values.

**Figure 10 sensors-20-05606-f010:**
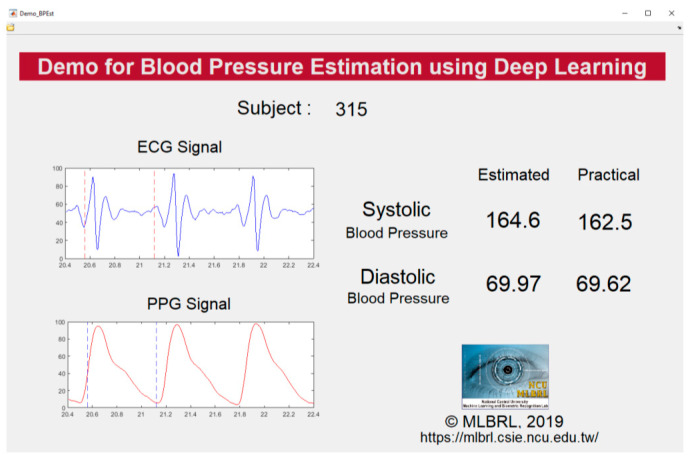
Our real-time demonstration program.

**Figure 11 sensors-20-05606-f011:**
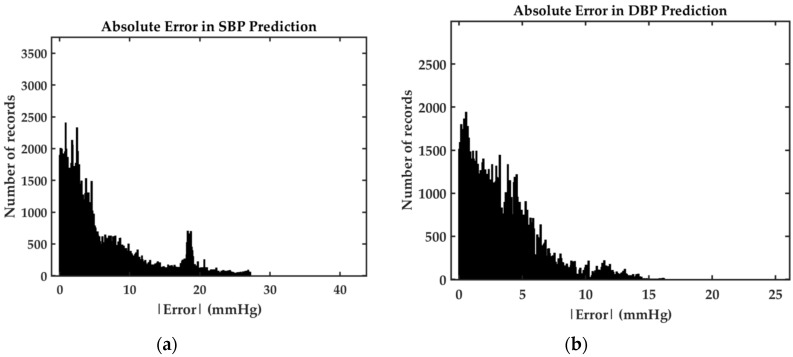
(**a**) SBP absolute error histogram from Model 2 (**b**) DBP absolute error histogram from Model 2.

**Table 1 sensors-20-05606-t001:** Features extracted from the ECG and the PPG signal, together with their unit.

#	Name of Features	Calculation	Unit
1	Pulse Transit Time (PTT)	Time interval between ECG R peak and the maximum slope of PPG signal in the same heart cycle.	second
2	Heart Rate (HR)	60 (s/min) divided by the time interval (s) between the ECG R peaks.	beats per minute
3	Reflection Index (RI)	$\frac{b}{a}$	arbitrary unit
4	Systolic Timespan (ST)	$tn_{n}-tf_{n}$	second
5	Up Time (UT)	$tp_{n}-tf_{n}$	second
6	Systolic Volume (SV)	$\overset{tn_{n}}{\underset{tf_{n}}{\int }}PPG\left(t\right)\;dt$	arbitrary unit
7	Diastolic Volume (DV)	$\overset{tf_{n+1}}{\underset{tn_{n}}{\int }}PPG\left(t\right)\;dt$	arbitrary unit

**Table 2 sensors-20-05606-t002:** Statistics of the blood pressure for the database used in the experiments.

	Min (mmHg)	Max (mmHg)	STD (mmHg)	Mean (mmHg)
**DBP**	60.00	129.97	9.03	70.42
**MAP**	66.82	145.18	10.39	91.72
**SBP**	80.00	179.99	19.74	134.33

**Table 3 sensors-20-05606-t003:** The proposed models compared with traditional machine learning methods on 1370 records.

Methods	SBP (mmHg)			DBP (mmHg)		
	MAE	STD	RMSE	MAE	STD	RMSE
Linear Regression [44]	9.1437	11.4959	10.5762	2.9791	1.2119	3.1675
Random Forest (Bag) [44]	2.6001	3.3633	2.9176	3.0228	1.3920	3.9702
LS Boost	4.8681	6.6808	5.4428	3.4522	1.8074	4.2513
Model 1 (4 LSTM)	1.1658	1.4003	1.5357	0.7475	0.8301	0.9877
Model 2 all connected (4 LSTM)	0.7357	0.9579	0.9379	0.5587	0.5088	0.6829
Model 3 all connected (5 LSTM)	1.7938	0.8070	1.9527	0.7469	0.4818	0.8563
Model 4 all connected (6 LSTM)	1.2405	0.7327	1.4323	1.0257	0.4553	1.0889

**Table 4 sensors-20-05606-t004:** Performance evaluation based on BP estimation standards on 135,641 records.

		Cumulative Error			ME (mmHg)	STD (mmHg)
		≤5 mmHg	≤10 mmHg	≤10 mmHg		
BHS [48]	Grade A	60%	85%	95%	-	-
	Grade B	50%	75%	90%	-	-
	Grade C	40%	65%	85%	-	-
AAMI [49]		-	-	-	< 5	< 8
Model 2	SBP	59.46%	79.97%	88.45%	4.638	14.505
	DBP	76.95%	95.72%	99.97%	3.155	6.442

**Table 5 sensors-20-05606-t005:** Our proposed model compared with deep learning methods on 135,641 records.

Methods	SBP (mmHg)			DBP (mmHg)		
	MAE	STD	RMSE	MAE	STD	RMSE
ANN-LSTM (s = 10) [7]	16.522	15.987	20.921	7.168	7.036	9.412
ANN-LSTM (s = 32) [7]	8.265	8.375	11.075	2.533	1.909	4.861
DNN	11.671	4.249	13.905	14.213	1.111	13.656
Model 2	6.726	14.505	8.051	2.516	6442	3.998

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
