# Peer review of "Real-Time Cuffless Continuous Blood Pressure Estimation Using Deep Learning Model"

_sensors, 2020, doi:10.3390/s20195606_

Round 1

Reviewer 1 Report

This manuscript presents an application of deep learning in continuous blood pressure measurement. The authors used the LSMT algorithm to deal with the nonlinear mapping of physiological parameters to blood pressure value. This is indeed the trend in medical monitor nowadays. However, it is necessary to deal with some improvements.

  1. There is too long redundant part in the manuscript. The first 10 pages are all about the existing work. There is a long section discussing the well-known basics which is of a low academic standard. There is no need to go into too much detail about the basics, such as the well-known MAE, RMSE, STD definitions and FFT method.

  1. “The conventional cuff-based BP measurement devices is discontinuous in nature.” The manuscript cites reference 4 in Line 46 and claims that the traditional method is discontinuous. However, the reference 4 also seems to be a continuous blood pressure monitoring method based on LSMT. Is this citation surely correct? What are the specific differences between this manuscript and reference 4?

  1. In addition, the selection of 7 input parameters in Line 179 seems to be highly similar to that in reference 25, but it is not declared. Is this a very common way of selecting parameters? If not, it needs to be explained. And what are the specific differences between this manuscript and reference 25?

  1. What does the “arbitrary unit” mean in Line 179 Table 1, dimensionless or what? Why does the volume have an “arbitrary unit”?

  1. The text size of the axis labels in the picture should be the same as the text of the manuscript. The text in Figure 1, 9 and 10 is not clear at all.

  1. In Line 357 the authors state that ‘remove some parts with very high or very low BP values’ What’s the reason makes it to be very high or very low? Is the extreme value supposed to be this or caused by calculation? If that’s the actual data, then why do you want to remove it.

  1. In Line 363 the abbreviation ‘STD’ is first mentioned. Please give its full name and check for similar cases.

  1. The conclusion is too simple and not convincing enough. Have the so-called other traditional machine learning methods used for comparison in Table 3 reached the upper limit of this algorithm? If possible, it is better to compare the results with existing work, or to make an analysis based on blood pressure monitor standard.

  1. Because of the uniqueness of the human body, the blood pressure calculation model will not be exactly the same for everyone. The overall error can only reflect part of the problem, and the distribution of the error is also important to evaluate the adaptability and robustness of the algorithm. So it's better to have some analysis about the distribution of the error.

  1. Line 153, missing “.”.

Author Response

Dear Editor and Reviewer

    We would like to thank you for valuable comments and suggestions. We have revised our manuscript according to the comments raised by the reviewer. The replied details have been highlighted in red color and are shown as follows.

Point 1: There is too long redundant part in the manuscript. The first 10 pages are all about the existing work. There is a long section discussing the well-known basics which is of a low academic standard. There is no need to go into too much detail about the basics, such as the well-known MAE, RMSE, STD definitions and FFT method.

Response 1: Thank you so much for the suggestion. We have modified the manuscript according to your opinion and remove the descriptions of the details about basic metrics and algorithms. The descriptions related to error metrics can refer to lines 386-390 on page 15. We have also revised the description on FFT method, which can be referred to lines 143-159 on page 5 (originally with more lines on 139-164). The other contents have also been revised in some extent without loss the completeness of the article.

Point 2: “The conventional cuff-based BP measurement devices is discontinuous in nature.” The manuscript cites reference 4 in Line 46 and claims that the traditional method is discontinuous. However, the reference 4 also seems to be a continuous blood pressure monitoring method based on LSMT. Is this citation surely correct? What are the specific differences between this manuscript and reference 4?

Response 2: Thank you so much for the correction. The citation has been changed to the correct one (Geddes, L. A.; Voelz, M.; Combs, C.; Reiner, D.; Babbs, C. F., “Characterization of the oscillometric method for measuring indirect blood pressure”. Annals of Biomedical Engineering, 1982, vol.10, no.6, pp.271-280. Please refer to lines 500-501 on page 21). As for the previous citation’s method, it is based on the combination of ANN and LSTM. While our method is combination of bidirectional LSTM and the common unidirectional LSTM. Also, the previous citation extracted features without doing feature engineering. The related discussion can be referred to lines 450-460 on page 19.

Point 3:In addition, the selection of 7 input parameters in Line 179 seems to be highly similar to that in reference 25, but it is not declared. Is this a very common way of selecting parameters? If not, it needs to be explained. And what are the specific differences between this manuscript and reference 25?

Response 3: Thank you so much for reviewer’s comment. We have mentioned in lines 170-171 on page 6 that we use seven features that extracted in reference 26 as follows:

 “We follow the procedure described in [26] for features extraction. There are seven features to be used as the input to the deep learning models.”

Despite the same feature extraction, we applied it to different database and feed it into different deep learning model.

Point 4:What does the “arbitrary unit” mean in Line 179 Table 1, dimensionless or what? Why does the volume have an “arbitrary unit”?

Response 4: Thank you so much for the comment.The term “arbitrary unit” is commonly adopted in the literature to represent the relative unit of measurement for the ratio of amount from the same physical quantity (such as the feature RI in Table 1) or used to express the signal related to light intensity or spectral intensity in physiology (such as the features SV and DV in Table 1). For the former cases, as RI is derived from the ratio of amplitude, it is indeed a dimensionless quantity. For the latter cases, SV and DV are derived from the integration of PPG waveform in specific timing interval. PPG reflects the change of blood volume during the circulation, and it has been converted to electrical signal for convenient processing from its original optical form. Its amplitude may also deviate in different measurement module. For these reasons, the received PPG pattern is usually labelled by “arbitrary unit” as it cannot directly reflect the physiological implication (just like the blood pressure does) uniquely from its magnitude. For this reason, the integration is also labelled to be “arbitrary unit” in Table 1.

Point 5:The text size of the axis labels in the picture should be the same as the text of the manuscript. The text in Figure 1, 9 and 10 is not clear at all.

Response 5: Thank you so much for the suggestion. We have adjusted the pictures and modified the manuscript accordingly. In this revised version, we try our best to maintain the size of characters on figures to be of the same (or at least approximate) size to those shown in the context. For the mentioned figures, they are now very clear in view on the manuscript. Please refer to page 3, 13, and 16, respectively.

Point 6:In Line 357 the authors state that ‘remove some parts with very high or very low BP values’ What’s the reason makes it to be very high or very low? Is the extreme value supposed to be this or caused by calculation? If that’s the actual data, then why do you want to remove it.

Response 6: Thank you so much for the comment. Since the data used in this paper (MIMIC II dataset) is collected from intensive care units (ICU), the majority of the samples are influenced by drugs which could potentially cause abnormal BP variation. Due to this, we consider extreme values in this dataset as outliers that can harm our estimation results. This practice was also conducted by Kachuee et al., (2017), one of the highly cited paper in this domain area of study. This paper is adopted to be reference [44] in this article.

Point 7:In Line 363 the abbreviation ‘STD’ is first mentioned. Please give its full name and check for similar cases.

Response 7: Thank you so much for the suggestion. We have added the full name as suggested when the term STD is first mentioned in the manuscript. Please refer to line 369 on page 14.

Point 8:The conclusion is too simple and not convincing enough. Have the so-called other traditional machine learning methods used for comparison in Table 3 reached the upper limit of this algorithm? If possible, it is better to compare the results with existing work, or to make an analysis based on blood pressure monitor standard.

Response 8: Thank you so much for the suggestion. We have added Table 4 (please refer to line 431 on page 18) where we demonstrated the performance evaluation of our model (Model 2) based on two BP estimation standards, British Hypertension Society (BHS) standard and Association for the Advancement of Medical Instrumentation (AAMI) standard. The detailed explanation is given at lines 432-443 on page 18. We have also modified the conclusion accordingly (please refer to lines 471-487 on page 20).

Point 9:Because of the uniqueness of the human body, the blood pressure calculation model will not be exactly the same for everyone. The overall error can only reflect part of the problem, and the distribution of the error is also important to evaluate the adaptability and robustness of the algorithm. So it's better to have some analysis about the distribution of the error.

Response 9: Thank you so much for the suggestion. We have given our analysis about the distribution of the absolute error at lines 440-443 on page 18. We have also added the histogram of the absolute error in Figure 11 as suggested (please refer to line 448 on page 19).

Point 10:Line 153, missing “.”.

Response 10: Thank you so much for the suggestion. We have modified the context of the manuscript accordingly, please refer to lines 152-153 on page 5.

Reviewer 2 Report

The paper describes an approach for realtime monitoring of blood pressure using LSTM neural networks. The work is interesting
in a sense that I personally haven't seen LSTMs applied in this context before. However, I cannot recommend publication due the following reasons.

The paper is that it is basically represents a comparison of their LSTM based model to a few other regression methods that are applied to the same dataset (i.e. applying the same features). The numerical values of errors seems good, but it is hard to judge actual performance. The authors could compare results also previous studies. In addition, the set of baseline regression models is quite limited. They could also compare results to other deep learning models without memory. Now it it hard to say that long term memory in LSTM really needed.

More specific comments:

- Section 2: the validity of (4) and its derivate is debatable. First, PWV is not really constant, it can vary significantly (e.g. ~5m/s near aorta and ~10 m/s in distal arteries). Also, as authors mention, "some parameters are closely related to personal arterial characteristics". But on the other hand, it seems that the paper does not use (4) or it just makes to wonder that what is the meaning of this derivation.

- Section 3: PTT seem to be defined as time interval between R-peak and the point of the maximum slope or the maximum derivate at systolic period. Some authors may define it so, but this is not only choice and it is debatable if it the best one. Stricly speaking, PTT is defined as the time interval between aortic valve opening and arrival of the wave to distal location (which is the local minima in the pressure wave, not the maximum slope). The timing relative to R-peak of course commonly used surrogate and it is often referred as the pulse arrival time. These timing can be significantly different due to variations in the pre-ejection period (PEP, from R-peak to the aortic valve opening); see e.g. Balmer J, Pretty C, Davidson S, Desaive T, Kamoi S, Pironet A, et al. Pre-ejection period, the reason why the electrocardiogram Q-wave is an unreliable indicator of pulse wave initialization. Physiol Meas. 2018;39(9):095005.)
These should be commented in the manuscript.

- Section 3.1: this seems to be just zeroing lowest frequencies after FFT. This is quite commonly known, so this section can be significantly shortened, or even left out and replaced with few lines in section 4. On the other hand, the manuscript does not mention that are recordings always 4096 samples long or did they apply, for example, fft in windows?

- Section 3.2 & Figure 2: Calculation of "PTT" is explained, but others are not. For instance, 1) How tf_n is chosen? I guess it should be the minimum in the pulse waveform signal, but it seems to not be so (Figure 2). 2) How do you detect dicrotic notch? 3) I cannot see any peaks in "Second peak". Peak of the derivative?

- Section 3.2: SV and DU are integrals of PPG signal without any normalization. Unit of PPG signal is arbitrary and the value of the integrals can depend on the amplification/scaling of individual recordings. If amplification or scaling changes between records, how it is made sure that these values are meaningful?

- Section 4, Table 3 and 4: I would assume that the values of the errors are mmHg, but the range of the values are very different between Table 3 and Table 4. For example, MAE for Model 3 is 0.7357 in Table 4 but 6.726 in Table 3. The very big difference raises questions.

Minor comments:

- The writing could perhaps be improved, but it is mostly understandable. But the organisation of the manuscript could be improved. Some details are missing and some parts of texts are very long without any apparent reason (some examples below).

- The manuscript gives an impression that the approach would allow long term continuous monitoring. However, the approach relies on ECG signal which can be cumbersome to use in long term. Furthermore, their PPG signal is from finger tip sensor that are mostly used in clinical setting. Alternatively, one can use for instance a PPG sensor in a wrist device, but signal quality is not as good. This could be commented.

- p.2 70-77: "For BP estimation, the output layer of the network always consists of two neurons, one for SBP and the other for DBP. The metric used for the
74 performance evaluation of the model was mean absolute error (MAE)." This is written as it would be common choice, but no references given ([11] is a generic deep learning book). Is this authors suggestion or what is the basis of this?

Author Response

Dear Editor and Reviewer

    We would like to thank you for valuable comments and suggestions. We have revised our manuscript according to the comments raised by the reviewer. The replied details have been highlighted in red color and are shown as follows.

Point 1: The paper is that it is basically represents a comparison of their LSTM based model to a few other regression methods that are applied to the same dataset (i.e. applying the same features). The numerical values of errors seems good, but it is hard to judge actual performance. The authors could compare results also previous studies. In addition, the set of baseline regression models is quite limited. They could also compare results to other deep learning models without memory. Now it it hard to say that long term memory in LSTM really needed.

Response 1: Thank you so much for reviewer’s suggestion. We have conducted a comparative analysis with previous study (reference [7]) using the same training and testing set to make it fair. We also compare our model performance to a deep neural network model which does not have memory cell to judge whether the long term memory in LSTM is really needed. We have added Table 5 to present the performance results to the manuscript (please refer to line 469 on page 20).

Point 2:Section 2: the validity of (4) and its derivate is debatable. First, PWV is not really constant, it can vary significantly (e.g. ~5m/s near aorta and ~10 m/s in distal arteries). Also, as authors mention, "some parameters are closely related to personal arterial characteristics". But on the other hand, it seems that the paper does not use (4) or it just makes to wonder that what is the meaning of this derivation.

Response 2: Thank you so much for the comment. PWV is indeed not constant. The derivation of Eq. (4) is based on the model for central artery near the heart and simultaneously the assumption on the constant elastic modulus of the arterial wall (referred to lines 116-118 on page 3). As it can be observed from Eq. (4) that there exists a nonlinear relationship between pressure Pand PTT, which forms the basis of cuff-less BP estimation in previous researches (such as References [2], [3], [5-9]). We also adopt PTT as one feature for BP estimation in this article. The parameters α(related to arterial stiffness), r(arterial radius), h(arterial thickness) and E0(elastic modulus at 0 mmHg) in Eq. (4) are closely related to personal arterial characteristics. As the morphology of PPG signal reflects indirectly the arterial characteristics, we also adopt the features from PPG pattern (such as RI, ST, UT, SV and DV, referred to Table 1 on Page 7). As the blood pressure may also regulate the heartbeat interval via the so-called baroreflex in the physiological system, heart rate (HR) is also selected to be one of the features in this article. From the above description, we indeed adopted Eq. (4) in this article. In addition to PTT, we also use several extra features acquired from PPG pattern which indirectly reflect the arterial characteristics embedded in Eq. (4) for the estimation of BP.

Point 3:Section 3: PTT seem to be defined as time interval between R-peak and the point of the maximum slope or the maximum derivate at systolic period. Some authors may define it so, but this is not only choice and it is debatable if it the best one. Strictly speaking, PTT is defined as the time interval between aortic valve opening and arrival of the wave to distal location (which is the local minima in the pressure wave, not the maximum slope). The timing relative to R-peak of course commonly used surrogate and it is often referred as the pulse arrival time. These timing can be significantly different due to variations in the pre-ejection period (PEP, from R-peak to the aortic valve opening); see e.g. Balmer J, Pretty C, Davidson S, Desaive T, Kamoi S, Pironet A, et al. Pre-ejection period, the reason why the electrocardiogram Q-wave is an unreliable indicator of pulse wave initialization. Physiol. Meas. 2018;39(9):095005.). These should be commented in the manuscript.

Response 3: Thank you so much for the comment. To avoid confusion, we have added the reviewer’s opinion in the article. The added content is as follows (which appears on lines 86-91 on page 2).

Strictly speaking, PTT is defined as the time interval between aortic valve opening and arrival of the blood flow to the distal location. The timing relative to the R-peak is a commonly used surrogate and is often referred as the pulse arrival time (PAT). Both timings can be significantly different due to the variations in pre-ejection period (PEP), which is from R-peak to the aortic valve opening [17]. This article follows the usage in previous literatures [3, 6, 16], the term PTT is also adopted to avoid confusion.

Point 4:Section 3.1: this seems to be just zeroing lowest frequencies after FFT. This is quite commonly known, so this section can be significantly shortened, or even left out and replaced with few lines in section 4. On the other hand, the manuscript does not mention that are recordings always 4096 samples long or did they apply, for example, fft in windows?

Response 4: Thank you so much for the comment. We have deleted some irrelevant content of section 3.1.1 in this revised version (please refer to page 5). In addition, the 4096 samples are used for the computation of FFT in each window to remove the low-frequency artifact in this article.

Point 5:Section 3.2 & Figure 2: Calculation of "PTT" is explained, but others are not. For instance, 1) How tf_n is chosen? I guess it should be the minimum in the pulse waveform signal, but it seems to not be so (Figure 2). 2) How do you detect dicrotic notch? 3) I cannot see any peaks in "Second peak". Peak of the derivative?

Response 5: Thank you so much for the suggestion. In this research, we take the minimum of the waveform in the region as tf_n. We take the minimum of the subtraction between the signal and the straight line going from the systolic peak to the lowermost as tn_n which denotes the dicrotic notch. Lastly, the second peak is defined to be the minimum of the second derivative of the waveform following the dicrotic notch. We have modified the manuscript accordingly (please refer to lines 183-188 on page 6).

Point 6:Section 3.2: SV and DU are integrals of PPG signal without any normalization. Unit of PPG signal is arbitrary and the value of the integrals can depend on the amplification/scaling of individual recordings. If amplification or scaling changes between records, how it is made sure that these values are meaningful?

Response 6: Thank you very much for reviewer’s comment. We have conducted the normalization after the removal of low-frequency artifact. To make the content clearer, we have added the brief explanation of the procedure in Section 3.1.2 (please refer to lines 160-168 on page 6).

Point 7:Section 4, Table 3 and 4: I would assume that the values of the errors are mmHg, but the range of the values are very different between Table 3 and Table 4. For example, MAE for Model 3 is 0.7357 in Table 4 but 6.726 in Table 3. The very big difference raises questions.

Response 7: Thank you so much for reviewer’s comment. The big difference is due to the amount of data being used. Since the experiment in Table 3 used traditional machine learning as comparison, using an enormous data will be time consuming. Thus, we use smaller set (1,370 records for testing) from the same source. While the testing set used for deep learning models comparison comprises 135,641 records.

Point 8:The writing could perhaps be improved, but it is mostly understandable. But the organisation of the manuscript could be improved. Some details are missing and some parts of texts are very long without any apparent reason (some examples below).

- The manuscript gives an impression that the approach would allow long term continuous monitoring. However, the approach relies on ECG signal which can be cumbersome to use in long term. Furthermore, their PPG signal is from finger tip sensor that are mostly used in clinical setting. Alternatively, one can use for instance a PPG sensor in a wrist device, but signal quality is not as good. This could be commented.

- p.2 70-77: "For BP estimation, the output layer of the network always consists of two neurons, one for SBP and the other for DBP. The metric used for the 74 performance evaluation of the model was mean absolute error (MAE)." This is written as it would be common choice, but no references given ([11] is a generic deep learning book). Is this authors suggestion or what is the basis of this?

Response 8: Thank you so much for the suggestion.

(1) We have modified the manuscript in the conclusion part according to the comment, which appears at lines 480-484 on page 20. The content is as follows:

“The dataset we used in this study acquire PPG from finger-tip sensor that are mostly used in clinical setting only. For more efficient BP monitoring, a wearable device with a PPG sensor on wrist can alternatively be used. However, the signal contains a lot more noise compared with thePPG obtained from finger-tip sensor.”

(2) MAE is one of the common metrics used to evaluate results of the prediction in regression task such as BP estimation. We have cited some prior researches dealing with BP estimation that used MAE as the evaluation metrics in the manuscript. Please refer to line 74 on page 2.

Round 2

Reviewer 1 Report

After a lot of modification, the research content of the manuscript look good. And I think it has met publishing standards. The only thing is that the manuscript is too long. I hope the author will pay attention to this problem.

Author Response

Dear Editor and Reviewer

    We would like to thank you for valuable comments and suggestions till now. We have revised our manuscript according to your comments. The replied details have been highlighted in red color and are given as follows.

Comment: After a lot of modification, the research content of the manuscript look good. And I think it has met publishing standards. The only thing is that the manuscript is too long. I hope the author will pay attention to this problem.

Reply: Thank you so much for the comment. We have revised our manuscript to make it shorter, from the previous 23 pages shorten to the present 21 pages.

Reviewer 2 Report

The authors have have mostly revised the paper sufficiently. But there are still few points I am not fully satisfied. The "points" below refer to the points in their cover letter.

Point 3: Authors could have used their own words in their manuscript, instead of just copying the reviewer's response almost word-by-word.

Point 5: The added description is sufficient. But the point for the foot tf_n in Fig. 2 is not in the minimum point. This can be confusing.

Point 7: This point raises even more questions. It is not normal that the range of values changes 10x when considering a bigger set instead of a small subset(?). The small set is badly chosen? And please add "mmHg" as unit to all of the tables.

Point 8: Sorry, I was not clear enough in my earlier comment. My comment was pointing to the sentence "For BP estimation, the output layer of the network always consists of two neurons, one for SBP and the other for DBP. " This sentence gives an impression that the output of the neural net is always SBP and DBP (there is really freedom of choice).

Author Response

Dear Editor and Reviewer

    We would like to thank you for valuable comments and suggestions till now. We have revised our manuscript according to your comments. The replied details have been highlighted in red color and are given as follows.

    The authors have mostly revised the paper sufficiently. But there are still few points I am not fully satisfied. The "points" below refer to the points in their cover letter.

Point 3:Authors could have used their own words in their manuscript, instead of just copying the reviewer's response almost word-by-word.

Response 3: Thank you so much for the suggestion. We have revised the manuscript accordingly (please refer to lines 87-93 on page 2), as follows:

“In practical condition, the term PTT refers to the travel time between aortic valve opening and arrival of the blood flow to the distal location. When the time is measured relative to the ECG QRS complex then it is generally used to define the term pulse arrival time (PAT), an interchangeably measure of PTT. Despite that, both timings could implicate poor correlation due to the variability of the pre-ejection period (PEP), which is from the related ECG QRS complex to the aortic valve opening [17]. This article follows the usage in previous literatures [3, 6, 16]and the term PTT is also adopted to avoid confusion.”

Point 5:The added description is sufficient. But the point for the foot tf_n in Fig. 2 is not in the minimum point. This can be confusing.

Response 5:Thank you so much for reviewer’s suggestion. We have adjusted the figure accordingly. Please refer to lines 142-143 on page 5.

Point 7:This point raises even more questions. It is not normal that the range of values changes 10x when considering a bigger set instead of a small subset(?). The small set is badly chosen? And please add "mmHg" as unit to all of the tables.

Response 7:Thank you so much for the comment and suggestion. We would like to point out that in our tests we got smaller error when we use smaller dataset and bigger error when using bigger dataset. Taking SBP estimation as an example, the best MAE on 1,370 number of records is 0.7357 mmHg (refer to Table 3 on page 16), while the best MAE on 135,641 records is 6.762 mmHg (refer to Table 5 on page 18). We consider the rise of error in bigger dataset is due to the increase of the variability in the data.

In addition, we have added “mmHg” as unit to the tables. Please refer to line 422 on page 16 and line 467 on page 18.

Point 8:Sorry, I was not clear enough in my earlier comment. My comment was pointing to the sentence "For BP estimation, the output layer of the network always consists of two neurons, one for SBP and the other for DBP. " This sentence gives an impression that the output of the neural net is always SBP and DBP (there is really freedom of choice).

Response 8:Thank you so much for reviewer’s comment. We admit that the output of the model is really freedom of choice. Therefore, we have added some citations that used two neurons as the output and changed the manuscript as follows (please refer to line 72-73 on page 2):

“For BP estimation, the output layer of the network often consists of two neurons, one for SBP and the other for DBP [7, 12-14].”
